# Cytokines and Chemokines Involved in Hepatitis B Surface Antigen Loss in Human Immunodeficiency Virus/Hepatitis B Virus Coinfected Patients

**DOI:** 10.3390/jcm10040833

**Published:** 2021-02-18

**Authors:** Noboru Urata, Tsunamasa Watanabe, Noboru Hirashima, Yoshiyuki Yokomaku, Junji Imamura, Yasumasa Iwatani, Masaaki Shimada, Yasuhito Tanaka

**Affiliations:** 1Department of Gastroenterology, National Hospital Organization, Nagoya Medical Center, Nagoya 460-0001, Japan; urata.noboru.df@mail.hosp.go.jp (N.U.); hirashima.noboru.pf@mail.hosp.go.jp (N.H.); shimada.masaaki.za@mail.hosp.go.jp (M.S.); 2Department of Virology and Liver Unit, Graduate School of Medical Sciences, Nagoya City University, Nagoya 467-8602, Japan; 3Division of Gastroenterology and Hepatology, Department of Internal Medicine, St. Marianna University School of Medicine, Kawasaki 216-8511, Japan; twatanab@marianna-u.ac.jp; 4Department of Infectious Diseases and Immunology, National Hospital Organization, Nagoya Medical Center, Nagoya 460-0001, Japan; yokomaku.yoshiyuki.zd@mail.hosp.go.jp (Y.Y.); imamura.junji.ua@mail.hosp.go.jp (J.I.); iwatani.yasumasa.cp@mail.hosp.go.jp (Y.I.); 5Department of Infectious Diseases and Immunology, National Hospital Organization, Sendai Medical Center, Sendai 983-8520, Japan; 6Department of Gastroenterology and Hepatology, Faculty of Life Sciences, Kumamoto University, Kumamoto 860-8556, Japan

**Keywords:** hepatitis B virus, human immunodeficiency virus, immune reconstitution syndrome, HBsAg, cytokine, chemokine

## Abstract

It has been reported that hepatic flare (HF), attributable to the development of immune reconstitution inflammatory syndrome (IRIS) in human immunodeficiency virus (HIV)/hepatitis B virus (HBV) coinfected patients, occurs frequently after the start of anti-retroviral therapy (ART). We have observed several cases of hepatitis B surface antigen (HBsAg) loss after IRIS. However, the factors leading to HBsAg clearance remain unknown. We measured CD4+ and CD8+ T cells, cytokines and chemokines in 16 patients coinfected HIV-1 and HBV with IRIS, and analyzed the factors leading to HBsAg clearance after IRIS. There was no significant difference in the CD4+ and CD8+ T cell counts between the HBsAg clearance and non-clearance groups, while the serum concentrations of almost all cytokines and chemokines in the HBsAg clearance group were higher than in the HBsAg non-clearance group at any time of observation. In particular, IP-10 at the ALT peak, GM-CSF and IL-12 one month after the ALT peak and TNF-α and GM-CSF after the ALT concentrations fell to within normal limits, were significantly higher in the HBsAg clearance group. It seems that HBsAg loss after IRIS requires continued immune responses against HBV, involving Th1 cytokines.

## 1. Introduction

It is estimated that there are approximately 39.0 million people living with human immunodeficiency virus (HIV) and 30 thousand of them were living in Japan at the end of 2019 [1,2]. Approximately 7.4% of the HIV-infected population worldwide, and 5–8% of population in Japan, are infected with hepatitis B virus (HBV) [3,4]. Patients coinfected HIV-1 and HBV, especially those with low CD4+ counts, are at increased risk of liver-related mortality [5]. Initiation of anti-retroviral therapy (ART) in HIV and HBV coinfected patients leads to a decline in HIV RNA concentrations improvement in CD4+ T cell counts and partial restoration of overall immune function, but recovery of cell-mediated immunity may elicit an immune response to HBV, leading to exacerbation of hepatitis [6,7]. This phenomenon is known as “Immune Reconstitution Inflammatory Syndrome (IRIS)” and, although there is no agreed definition of IRIS, several definitions have been advocated. IRIS has been reported in 10–32% of patients starting ART [6]. IRIS can cause an increase in alanine aminotransferase (ALT) levels or hepatic flare (HF) [8]; HF being defined as an ALT level >5 times the upper limit of normal or >100 U/L higher than that observed at baseline [9]. HFs are seen in 20–25% of patients after initiation of ART, an estimated 1–5% of whom develop clinical hepatitis [10]. We have observed some cases of hepatitis B surface antigen (HBsAg) loss, i.e., functional cure of HBV infection, after HBV-related IRIS. However, the factors associated with HBsAg clearance remain unknown. In this study, we measured CD4+ and CD8+ T cells, cytokines and chemokines in patients coinfected HIV-1 and HBV with IRIS, and analyzed the factors associated with HBsAg clearance after IRIS.

## 2. Materials and Methods

### 2.1. Patients

Between 2004 and 2010, 20 patients coinfected HIV-1 and HBV with IRIS after initiating ART in the Department of Infectious Diseases and Immunology, Clinical Research Center, National Hospital Organization Nagoya Medical Center, Nagoya, were enrolled in this study. However, three patients who had no specific measurement of baseline HBsAg levels and one patient without serial plasma samples were excluded (Figure 1). Study patients visited our hospital at every month, and blood samples were taken and stocked at every visit. HBV DNA and HIV-1 RNA were collected every 3 months. All patients received the first therapy for HIV/HBV at our hospital, and had been HBsAg positive for ≥6 months before initiation of ART. Finally, 16 patients with IRIS after initiating ART were classified according to the presence or absence of HBsAg loss during observation, and the group with HBsAg loss was designated as “HBsAg clearance group (*n* = 8)” and the group without HBsAg loss was designated as “HBsAg non-clearance group (*n* = 8)”.

### 2.2. Measurement of HIV-1 RNA and HBV DNA

HIV-1 RNA was assayed by real-time polymerase chain reaction (PCR) with a fully automated COBAS Ampliprep/TaqMan assay (Roche Diagnostics Corp., Rotkreuz, Switzerland). HBV DNA was assayed by real-time PCR with a fully automated COBAS 8800 system (Roche Diagnostics Corp., Rotkreuz, Switzerland). The results of HIV-1 RNA and HBV DNA were expressed as copies/mL and Log_10_ copies/mL.

### 2.3. Laboratory Tests and Measurement of Cytokines/Chemokines

We measured CD4+ and CD8+ T cells, cytokines and chemokines at six time points: at baseline before initiating ART, when serum ALT levels started to rise, when ALT reached peak, one month after the ALT peak, three months after the ALT peak and when ALT improved to within the normal range during IRIS. A Bio-Plex 200 system (Bio-Rad Laboratories, Hercules, CA, United States) was used for the simultaneous measurement of multiple targets. We comprehensively measured chemokines and cytokines that have the potential to induce inflammation and eliminate viruses. This system enabled screening of the expression of the following: interleukin-1b (IL-1b), interleukin-2 (IL-2), interleukin-4 (IL-4), interleukin-5 (IL-5), interleukin-6 (IL-6), interleukin-8 (IL-8), interleukin-10 (IL-10), interleukin-12 (IL-12 p70), interleukin-13 (IL-13), granulocyte macrophage colony-stimulating factor (GM-CSF), IFN-γ, IP-10, tumor necrosis factor-α (TNF-α), interleukin- 17A (IL-17A) and interleukin-21 (IL-21).

### 2.4. Ethical Considerations

The Ethical Review Committee of National Hospital Organization Nagoya Medical Center approved this study on 5 June 2012 (approval number 2012-503). This study was conducted in accordance with the guidelines set out by the Declaration of Helsinki. Informed consent was obtained from all individual participants included in this study.

### 2.5. Statistical Analysis

The numerical results are expressed as medians (minimum–maximum). Categorical variables were compared between groups using Man Whitney’s *U*-test. A value of *p* < 0.05 was considered statistically significant. Statistical analyses were performed using IBM SPSS Statistics version 24 (International Business Machines Corp., Armonk, NY, USA).

## 3. Results

### 3.1. Patients’ Characteristics

The characteristics of the 16 patients who participated in this study (8 in the HBsAg clearance group and 8 in the HBsAg non-clearance group) are presented in Table 1. The median age of patients was 34.5 years, and all were men who have sex with men (MSM). The median ALT was 43 U/L. The median HBsAg value was 125,000 IU/mL and all were HBeAg positive and HBeAb negative. The duration of HBsAg positivity in the 8 patients in the clearance group was 10.1 ± 4.2 months after ART initiation. The median HBV DNA value was 6.9 Log copies/mL. The median CD4+ T cell count was 150/µL and the median HIV-1 RNA value was 23,050 copies/mL. There was no significant difference between the HBsAg clearance group and HBsAg non-clearance group. There were 4 AIDS cases, including 3 HBsAg clearance cases. AIDS-defining illness was extrapulmonary tuberculosis, cytomegalovirus infection, pneumocystis pneumonia, candidiasis. The drugs used in ART were tenofovir disoproxil fumarate (TDF)/emtricitabine (FTC) + efavirenz (EFV) in 7 cases, TDF/FTC + darunavir/ritonavir (DRV/r) in 6 cases, TDF/FTC + raltegravir (RAL) in 1 case, and abacavir (ABC)/lamivudine (3TC) + atazanavir (ATV) in 1 case. Three patients have been lost to follow-up due to relocation, but we have been following up for a median of 11 years and 10.5 months (3 years and 9 months–14 years). We show two representative cases as follows.

Case 1 (HBsAg clearance group): A 36-year-old Japanese man who had sex with men was diagnosed with HIV-1 and HBV (Genotype A) in 2009 and treated initially with ART, TDF/FTC and DRV/r. Before the initiation of ART, his ALT levels were within the normal limits (32 U/L) and HBsAg was positive (125,000 IU/mL). HIV-1 infection was confirmed by western blotting, with a viral load 7200 HIV RNA copies/mL. One month after the initiation of ART, HIV RNA declined to 63.1 copies/mL, while his ALT levels rose to 440 U/L. One year later after the initiation of ART, he tested negative for HBsAg (Figure 2a).

Case 2 (HBsAg non-clearance group): A 33-year-old Japanese man who had sex with men was diagnosed with HIV-1 and HBV (Genotype A) in 2008 and treated initially with ART, TDF/FTC and EFV. Before the initiation of ART, his ALT level was within the normal limits (44 U/L) and HBsAg was positive (54,856 IU/mL). HIV-1 infection was confirmed by western blotting (viral load 220,000 HIV RNA copies/mL). One month after the initiation of ART, HIV RNA declined to 800 copies/mL, while his ALT levels rose to 90 U/L. HBsAg declined to 28,986 IU/mL one month later, however HBsAg was remained positive to 2020/11/12. (Figure 2b).

### 3.2. The Serum ALT Levels in the HBsAg Clearance and Non-Clearance Groups

The median serum ALT level at peak ALT in the HBsAg clearance group 482 (110–1255) U/L tended to be higher than the median in the non-clearance group 218 (90–475) U/L (*p* = 0.066). Thirteen patients presented with HF, 7 in HBsAg clearance group and 6 in HBsAg non-clearance group. The term from ART initiation to peak ALT was a median of 3 (1–6) months in the HBsAg clearance group and 3 (1–8) months in the HBsAg non-clearance group, with no significant difference (*p* = 0.457).

### 3.3. CD4+ and CD8+ T Cell Counts in the HBsAg Clearance and Non-Clearance Groups

After the initiation of ART, CD4+ and CD8+ T cell counts recovered and there was no significant difference between the HBsAg clearance group and HBsAg non-clearance group (Figure 3).

### 3.4. HBV DNA and HIV-1 RNA in the HBsAg Clearance and Non-Clearance Groups

HBV DNA declined more rapidly in the HBsAg clearance group. The number of cases in which HBV DNA was not detected gradually increased in HBsAg clearance group, with a trend toward more cases three months after peak ALT (*p* = 0.063) in the HBsAg clearance group, so there were significant differences of undetectable HBV DNA at the time point after ALT returned to the normal range (*p* < 0.05). The HBV DNA negative rate was 37.5% three months after ALT peak and 100% after ALT normalization, but 0% in the HBsAg non-clearance group. HIV-1 RNA decreased sharply in both groups by the time IRIS occurred, and there was no significant difference (Figure 4).

### 3.5. Cytokines and Chemokines in the HBsAg Clearance and Non-Clearance Groups

Regarding the HBsAg clearance and non-clearance groups, at the peak of ALT, during the acute hepatitis caused by IRIS, there was a significant difference in IP-10 (*p* = 0.047) but no significant difference in the other cytokines and chemokines (Figure 5).

One month after the ALT peak, there were significant differences in IL-12 p70 (*p* = 0.027) and GM-CSF (*p* = 0.044) concentrations, as well as a tendency for differences in IL-2 (*p* = 0.093), IL-8 (*p* = 0.081) and IL-10 (*p* = 0.071) (Figure 6).

After the ALT had returned to within normal limits, there were significant differences in GM-CSF (*p* = 0.033) and TNF-α (*p* = 0.040) concentrations, as well as a tendency in for a difference in IL-6 (*p* = 0.056) (Figure 7).

### 3.6. Changes in the Longitudinal Profiles of the Cytokines and Chemokines

IL-2 and IL-6, IL-8, IL-10, IL-12, GM-CSF, TNF-α were at overall higher levels in the HBsAg clearance group than the non-clearance group during IRIS. As for IP-10, there was a significant difference between the groups at the ALT peak. GM-CSF and IL-12 p70 differed one month after the ALT peak, as did TNF-α and GM-CSF after the ALT levels had returned to within normal limits (Appendix A).

## 4. Discussion

IRIS in HIV patients leads to improvements in CD4+ and CD8+ T cell counts and boosting of cellular immunity [6]. Some HIV and HBV coinfected patients with IRIS experience HF, and a proportion of them results in preferable clinical course as HBV functional cure. Therefore, we hypothesize that restoration of overall immune function leads to clearance of HBV and loss of HBsAg in HIV and HBV coinfected patients, and analysis of the immune response, i.e., cytokines and chemokines, could contribute to the future actions required to achieve HBV functional cure. In this study, there were no significant differences in CD4+ and CD8+ T cells counts between the HBsAg clearance and non-clearance groups. There have been studies looking at the immunobiology of the liver microenvironment in the HIV-HBV co-infected population after ART initiation [11]. A study of Iser et al. showed that among HIV-HBV coinfected patients on ART, a significant increase in the number of peripheral circulating CD4+ T cells but not in the liver. Intrahepatic changes by liver biopsy showed no significant changes in the number of CD4+ and CD8+ T cells or T cell activation, suggesting a low correlation with HBs antigen clearance [11]. On the other hand, there was a significant difference in IP-10 at the peak of ALT, during IRIS, as well as in IL-12 and GM-CSF one month after the ALT peak. Especially, IL-12 p70 which can stimulate NK cells, induce Th1 type cytokine production and foster the development of cytotoxic T lymphocytes, was released by dendritic cells (DCs) [12]. IL-12 induces the differentiation of naive CD4^+^ T cells into IFN-γ-producing Th1 cells [13], and it seems that IL-12 was associated with viral clearance in patients with HBV.

One month after the ALT peak during IRIS, and after the ALT levels had returned to within normal limits, GM-CSF was at high levels and induced and activated DCs, which served to break CTL immune tolerance and induce specific CTL reactions against HBV [12,14]. CTLs exert cytopathic effects, to kill a small fraction of the infected hepatocytes, and secrete IFN-γ and TNF-α, which exert noncytopathic antiviral effects without destruction of hepatocytes [15,16]. In fact, after ALT levels had returned to within normal limits, TNF-α was at high levels in the HBsAg clearance group.

In general, cytotoxic mechanisms for HBV-specific CD4+ and CD8+ T cells and noncytotoxic mechanisms for IFN-γ and TNF-α, which are produced by natural killer (NK) cells, are detectable in acute hepatitis B patients [17,18]. Activated NK cells can activate CD4+ and CD8+ T cells, DCs and NK T cells [19]. In acute HBV infection, a strong cell-medicated Th1 response, induced by IL-2 and IFN-γ, is mounted against HBV and is associated with viral clearance [20].

In this study, almost all cytokines and chemokines were at higher levels in the HBsAg clearance group than the non-clearance group, at any time of observation and it is suggested that, from the ALT peak to its return to within normal limits, continued inflammation involving immunoregulatory cytokines (IL-12) and pro-inflammatory cytokines (TNF-α, IL-8) and immune responses against HBV by Th1 cytokines, such as IL-2, are important for virus clearance. Siegbert et al. reported that the rise in serum IL-12 levels was associated with a significant increase in Th1 cytokines, and high levels of IFN-γ and IL-2 were observed [15]. Certain cytokine combination may induce a different effect on virus clearance. Then, our study has some limitations that should be taken into consideration; the number of cases is small, and the results should be verified with a large number of cases.

## 5. Conclusions

In HIV/HBV coinfected patients who developed HBV-related IRIS, almost all cytokines and chemokines were at higher levels in the HBsAg clearance group than the non-clearance group, at all times of observation. There were significant differences between the HBsAg clearance and non-clearance groups in IP-10, IL12 and GM-CSF. It seems that continued immune responses by Th1 cytokines against HBV are necessary for HBsAg loss after IRIS.

## Figures and Tables

**Figure 1 jcm-10-00833-f001:**
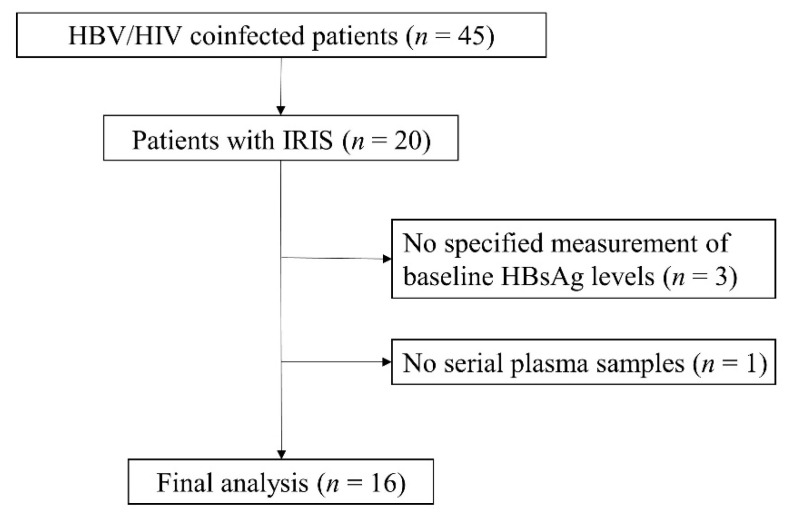
Study flow diagram from enrollment to the final analysis. Abbreviations: HBV, hepatitis B virus; HIV, human immunodeficiency virus; IRIS: immune reconstitution inflammatory syndrome; HBsAg, hepatitis B surface antigen.

**Figure 2 jcm-10-00833-f002:**
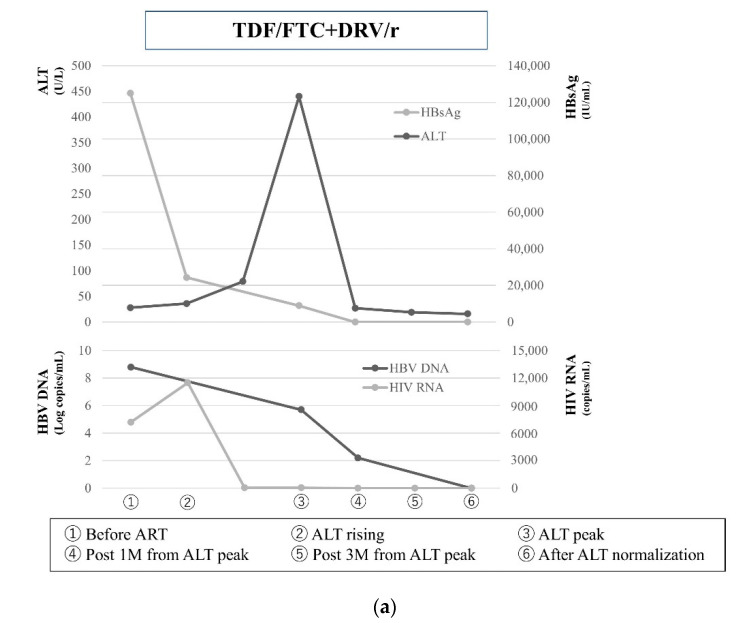
(**a**) A case from the HBsAg clearance group. He experienced hepatic flare and subsequent HBsAg loss after ART initiation. Abbreviations: TDF, tenofovir disoproxil fumarate; FTC, emtricitabine; DRV/r, darunavir/ritonavir. (**b**) A case from the HBsAg non-clearance group. He experienced HBV-related IRIS and not HBsAg loss after ART initiation. Abbreviations: TDF, tenofovir disoproxil fumarate; FTC, emtricitabine; EFV, efavirenz.

**Figure 3 jcm-10-00833-f003:**
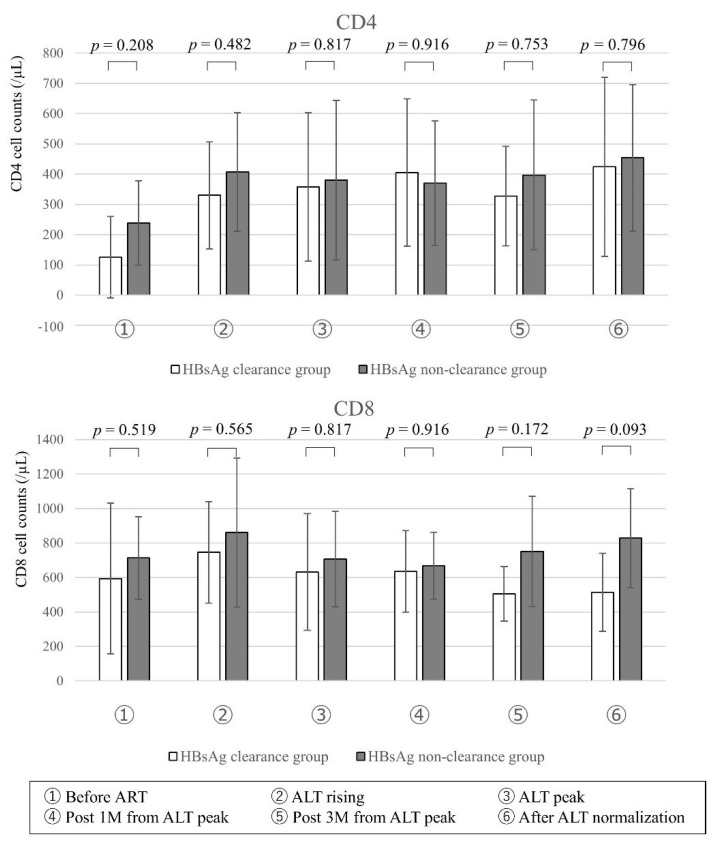
Change of CD4+ T cell and CD8+ T cell counts. CD4 and CD8 cell counts showing significantly different (*p* < 0.05 by the Mann Whitney *U*-test) between HBsAg clearance group (*n* = 8) and HBsAg non-clearance group (*n* = 8) are represented. Data are expressed as means ± standard deviation. Abbreviations: CD4, cluster of differentiation 4; CD8, cluster of differentiation 8.

**Figure 4 jcm-10-00833-f004:**
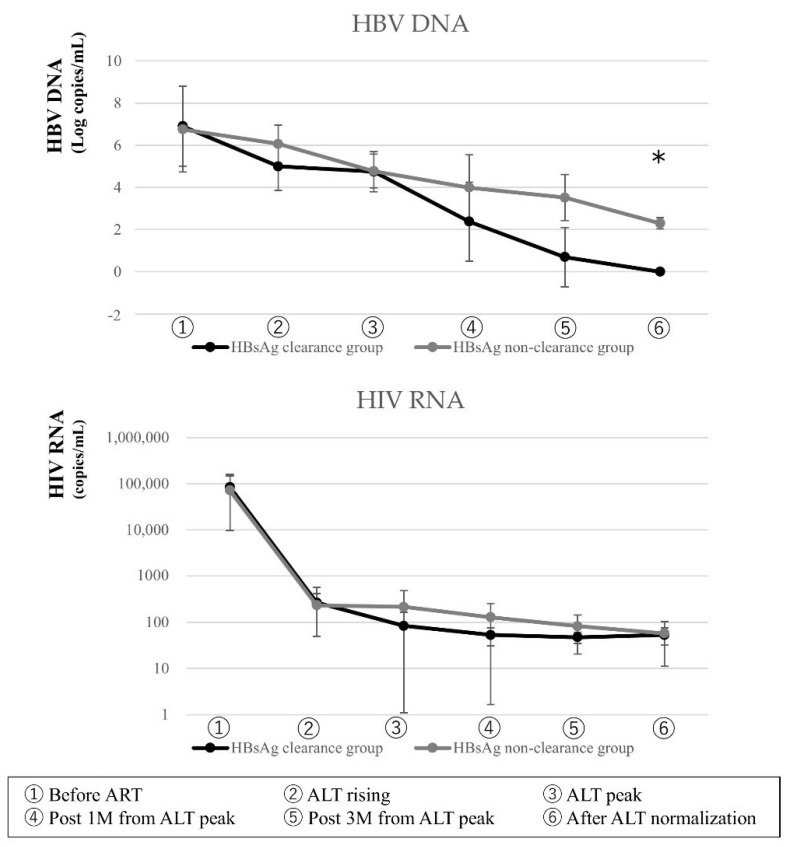
Change of HBV DNA and HIV-1RNA. HBV DNA and HIV-1 RNA showing significantly different (*p* < 0.05 by the Mann Whitney *U*-test) between HBsAg clearance group (*n* = 8) and HBsAg non-clearance group (*n* = 8) are represented. Data are expressed as means ± standard deviation. The symbol “*” indicates a *p*-value <0.05.

**Figure 5 jcm-10-00833-f005:**
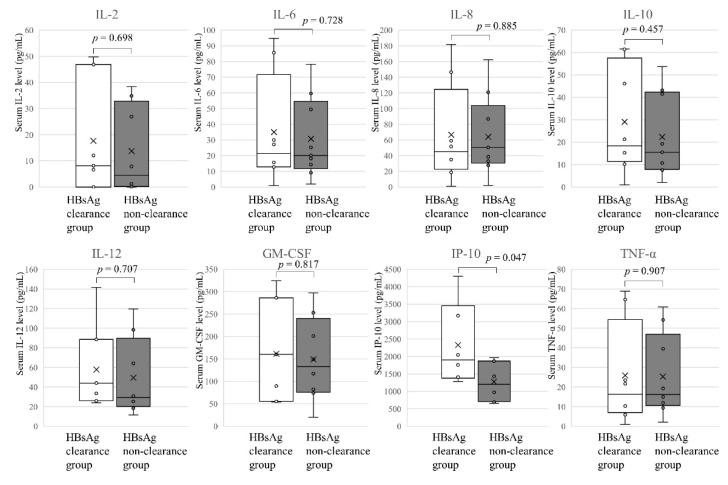
Cytokines and chemokines at the ALT peak. Eight cytokines and chemokines levels showing significantly different (*p* < 0.05 by the Mann Whitney *U*-test) between HBsAg clearance group (*n* = 8) and HBsAg non-clearance group (*n* = 8) are represented.

**Figure 6 jcm-10-00833-f006:**
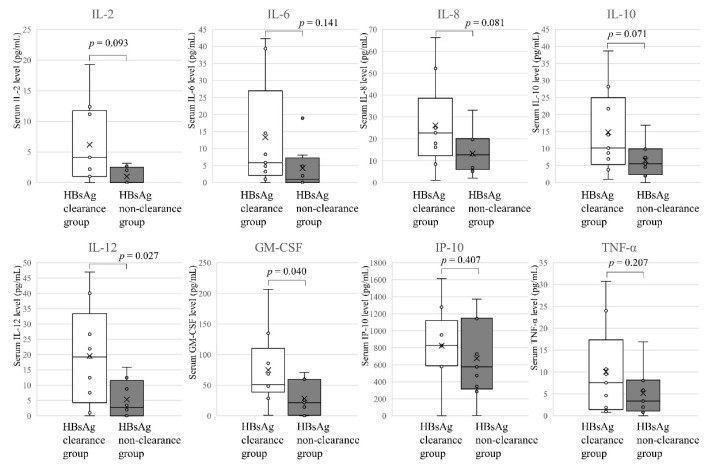
Cytokines and chemokines one month after the ALT peak. Eight cytokines and chemokines levels showing significantly different (*p* < 0.05 by the Mann Whitney *U*-test) between HBsAg clearance group (*n* = 8) and HBsAg non-clearance group (*n* = 8) are represented.

**Figure 7 jcm-10-00833-f007:**
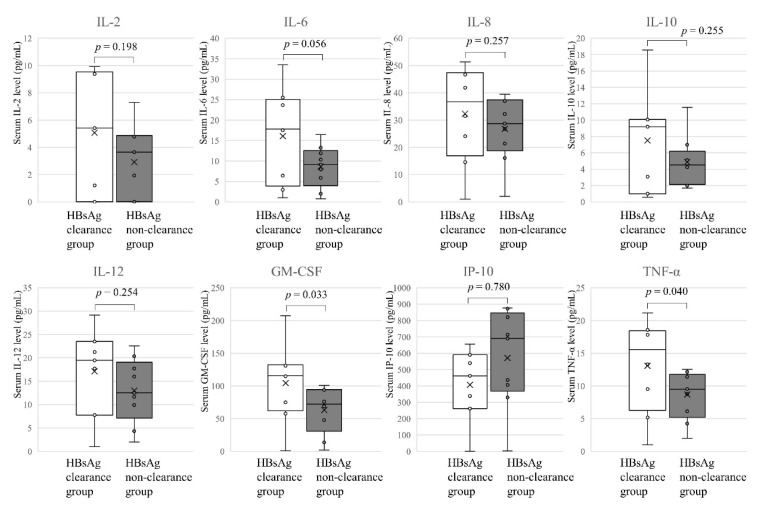
Cytokines and chemokines after the ALT levels returned to within normal limits. Eight cytokines and chemokines levels showing significantly different (*p* < 0.05 by the Mann Whitney *U*-test) between HBsAg clearance group (*n* = 8) and HBsAg non-clearance group (*n* = 8) are represented.

**Table 1 jcm-10-00833-t001:** Clinical Backgrounds of the Patients.

	All (*n* = 16)	HBsAg Clearance Group (*n* = 8)	HBsAg Non-Clearance Group (*n* = 8)	*p*
Age (years)	34.5 (29–63)	37 (29–63)	32.5 (29–58)	0.206
Sex	All male			
ALT (U/L)	43 (22–193)	45 (22–95)	43 (28–193)	0.793
HBsAg (IU/mL)	125,000 (23,327–125,000)	89,034 (23,327–125,000)	125,000 (40,550–125,000)	0.298
HBV DNA (Log_10_ copies/mL)	6.9 (3.3–9.1)	6.9 (4.0–9.1)	6.9 (3.3–8.8)	0.710
CD4 (/µL)	150 (3–458)	76.5 (3–458)	296.5 (10–391)	0.208
HIV-1 RNA (copies/mL)	23,050 (450–220,000)	51,550 (13,500–210,000)	23,000 (450–220,000)	0.833

Data are expressed as the median (minimum–maximum). Abbreviations: ALT, alanine aminotransferase; HBsAg, hepatitis B surface antigen; HBV, hepatitis B virus; CD4, cluster of differentiation 4; HIV, human immunodeficiency virus.

## Data Availability

Data sharing is not applicable to this article. The data are not publicly available due to restrictions of privacy and ethics.

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
