# Peer review of "Cytokines and Chemokines Involved in Hepatitis B Surface Antigen Loss in Human Immunodeficiency Virus/Hepatitis B Virus Coinfected Patients"

_jcm, 2021, doi:10.3390/jcm10040833_

Round 1
Reviewer 1 Report
1. Introduction
Line 55 - replace "experience" with "observed"
2. Materials and Methods
Line 69 - Authors categorized HBsAg clearance group with IRS which is not accurate. HBV-IRS does not necessarily mean patients will end HBsAg clearance. This should be clarified and changed.
Line 76 - Add space between "HIV-1" and "RNA".
Line 77 - change Log to Log10
Lines 82-88 describes the cytokines and chemokines assays used in this study. Are there HBV specific cytokines or just overall general nonspecific chemokines/cytokines assayed?
Other comments:
- Clinical studies that enrolls human subjects need statement that subjects gave necessary informed consent and that the study is approved by an IRB or similar ethics committee.
- It's not clear to me as to how often liver enzymes monitored after initiating ART. I think methodology should be much more clearly described in this section.
- How often was HIV RNA and HBV DNA monitored during study?
- Authors mentioned on Lines 79-81, that cytokines and chemokines were measured at six time points but not clear to me what it corresponds to with regards to onset of ART? When did the ALT peak in subjects who cleared HBsAg versus who didn't?
Figure 1 - Second box from top: what does "IRH" mean? IRS?
Figure 1 - Fourth box from top: correct "plasm" to "plasma"
3. Results
Table 1 - Include baseline HBV DNA
Table 1 - HBsAg range for all had lower limit of 23,327 IU/mL but for the HBsAg clearance group lower limit was 2,327 IU/mL. Please correct discrepancy.
Line 12-13. "HIV RNA declined to 63.1 copies/mL..." when? 1 month after ART initiation?
Other comments/recommendations
- Describe and compare the HIV RNA and HBV DNA viral kinetics between the HBsAg clearance and non-clearance groups.
- With time of ART initiation as reference point, describe or detail as to when do you expect ALT to peak in subjects and compare both groups.
- Was the ALT peak higher in HBsAg clearance group than the non-clearance group?
- With time of ART initiation as reference, when did subject who develop IRS clear HBsAg (for the 8 who did)?
Figures 2a and 2b - TVD is not the accepted abbreviation for tenofovir (either use TFV for tenofovir, TDF for tenofovir disoproxil fumarate, or TAF for tenofovir alafenamide)
Reviewer 2 Report
In their manuscript, Urata et al. studied the chemokine and cytokine levels after initiating ART in what might be HIV-positive individuals with chronic HBV infection, while comparing these levels between individuals who seemed to have IRIS-induced HBsAg-seroclearance versus no HBsAg-seroclearance. There were certain chemokines, namely the Th1 cytokines, that seemed to be higher in those with HBsAg-seroclearance. Interestingly, these levels remained higher even after peak ALT levels, which suggests that ongoing inflammation could be the reason for seroclearance. The study is on an very interesting topic and in line with much of the HBV cure research done currently, but there a number of issues that need to be addressed.
The most important concern is the imprecisions of defining the HBV infected group. The authors claim that this is acute HBV infection, but did they consecutively test HBsAg to determine HBsAg-positivity within a 6-month interval? And the population observed in the “HBsAg clearance group” are just individuals who cleared acute infection (and did not develop chronic infection)? Even so, if these individual are men who have sex with men (MSM) or persons who inject drugs (PWID), there could even be delayed seroconversion (van Santen et al, J Viral Hepat. 2020), making diagnosis of acute infection unclear.
Or are these individuals who had previous infection (i.e. HBsAg-negative, anti-HBs antibody positive, anti-HBc antibody positive) and after ART initiation, had reactivation of HBV infection? This seems to be suggested with wording: “…with acute hepatitis caused by IRS,…” in the abstract and the text or “acute exacerbations of chronic hepatitis B” (ln 165). The question becomes completely different then. Also, why would there be HBV reactivation if the patients are being treated with an anti-HBV containing ART? (See comments below.)
It seems more probable that these individuals had chronic HBV infection prior initiating ART and that accelerated HBsAg-seroclearance occurred after ART-initiation. (Did the authors confirm chronic HBV infection, i.e. two consecutive HBsAg-positive results >6 months? If not, this needs to be listed as a limitation.) The authors contend that this was due to an IRIS-like phenomenon (which has been suggested in multiple other studies: Miailhes P et al, Clin Inf Dis, 2007; Chihota BV et al, J Inf Dis, 2020; van Bremen K et al, Liver Int, 2020), yet given that many patients had relatively high CD4 cell counts (with 522/mm3 being the upper limit), the IRIS component can only partially explain these findings. All in all, the authors seem to demonstrate cytokine and chemokine differences in HIV-HBV co-infected patients who did versus did not have HBsAg-seroclearance immediately after ART initiation. The conclusions in regards to IRIS need to be reworded throughout the manuscript.
It is also unclear how patients were recruited. Were these all HIV-HBV patients entering care for the first time and initiated ART immediately? The proportion with HBsAg-seroclearance seems incredibly high compared to rates observed in Europe (Zoutendijk R et al, J Inf Dis, 2012), Africa (Boyd A et al, J Gastro Hepatol, 2015), or to a lesser extent, South-East Asia (Matthews GV et al, PLoS One, 2013). Is this population representative of HIV-HBV co-infected patients generally seen in Japan?
Furthermore, there is very little information regarding HIV in these patients. Any individuals with AIDS-defining illness? If so, which ones? What were the ARV agents used? And what proportion were undergoing ART with an anti-HBV agent? There is also missing information with regards to HBV infection. What was the HBV serological profile (i.e. HBeAg, anti-HBeAb, anti-HBc)? What time did HBsAg-seroclearance occur? And what were the HBV DNA viral loads prior to baseline and during follow-up? This is basic clinical information that needs to be included.
The timing of measurements is also puzzling. Instead of basing measurements at specific time points, the authors planned measurements at moments based on ALT measurements (e.g. during increase, at peak levels, and time periods after peak). How were the authors able to ascertain “peak” levels without some consistent testing intervals? And are the authors certain that they obtained levels at the true peak (given than ALT flares can last less than 7 days)? These are potential limitations worth discussing.
The statistical analysis could be improved. The authors could use a mixed-effect linear regression model (with random intercept to account for patient variability at ART-initiation), and model time, HBsAg-group and the interaction between time point and HBsAg group. The authors can perform statistical tests for differences at each time point and a joint test of differences across all time points as their “overall” test. If the authors are unable to perform these analysis, they should use a non-parametric test rather than the Student’s t-test.
Minor comments:
- Overall: The more accepted term is “immune reconstitution inflammatory syndrome” or IRIS; not IRS.
- ln 45-47. “worsening of a pre-existing condition”. This is mostly TB associated, not HBV. Reword.
- ln 50. How is diagnosing IRIS helpful? And how many individuals had IRIS in your study?
- ln 56. It is not “cure”, but “functional cure”.
- ln 68-69. Give the numbers of patients falling into these two categories.
- ln 92. “The symbol “*” indicates…” Not clear why the sentence is here. This seems to only apply to the figures. If so, it should only be included in the figure legends.
- Results. Did all patients have data at all visits during follow-up? It needs to be stated how long follow-up lasted for patients and how many individuals were lost to follow-up.
- ln 107-124. The case vignettes are mostly anecdotal and do not add anything to the manuscript. Delete.
- ln 128-131. “Provisionally, … as on group.” Also ln 137-140. This section is largely unnecessary. It should be listed as a limitation that you were unable to perform more granular analysis using qHBsAg levels at baseline.
- Figure 3. Please include either IQRs or some form of confidence bands.
- ln 185-190. There are some great studies looking at the immunobiology of the liver microenvironment in the HIV-HBV co-infected population after ART initiation (e.g., Iser DM et al, AIDS, 2011). These results need to be brought into context.
- Discussion. The authors need to discuss why there was no difference in CD4+ cell counts, especially as IRIS is the major hypothesis for HBsAg-seroclearance.
- Figures. All figures should include the numbers of individuals at each time point.
Round 2
Reviewer 1 Report
Minor grammatical edits needed:
ABSTRACT
Line 33 - Replace "experienced some" with "observed several"
Line 81 - Replace "at" with "collected". Sentence should read "HBV DNA and HIV-1 RNA were collected every 3 months."
Lines 136-141 - ART generic names should not be Capitalized unless located in the beginning of a sentence. For example, change "Tenofovir Disoproxil Fumarate" to "tenofovir disoproxil fumarate". Please correct throughout the manuscript.
Line 143 - add a space between "14" and "years"
Line 155 - remove "later"
Line 166 - remove "later"
Line 191 - remove "would" and change "decline" to "declined"
Line 193 - add space between "3" and month
Line 193 - replace "3" with "three"
Line 238 - replace "part" with "proportion"
Line 238 and 242 - Regarding "HBV cure": Complete HBV cure is currently not achievable, I would use the would HBV functional cure or HBsAg seroconversion.
Line 244 - remove "great"
Lines 246-250 - Better if "A study by Iser et al. showed that among HIV-HBV coinfected patients on ART, a significant increase in the number of peripheral circulating CD4+ T cells but not in the liver. Intrahepatic changes by liver biopsy showed no significant changes..."
Line 283 - simplify "HIV and HBV" to "HIV-HBV" or "HIV/HBV"
Author Response
We thank the reviewers for their insightful comments and suggestions, which have helped to improve our paper.
Reviewer 1 Comments and Suggestions for Authors
Minor grammatical edits needed:
ABSTRACT
Line 33 - Replace "experienced some" with "observed several"
Response: We thank you for your pointing out. According to your suggestion, the description "experienced some" was changed to "observed several" (line 33).
Line 81 - Replace "at" with "collected". Sentence should read "HBV DNA and HIV-1 RNA were collected every 3 months."
Response: We thank you for your pointing out. According to your suggestion, the description "at" was changed to "collected" (line 81).
Lines 136-141 - ART generic names should not be Capitalized unless located in the beginning of a sentence. For example, change "Tenofovir Disoproxil Fumarate" to "tenofovir disoproxil fumarate". Please correct throughout the manuscript.
Response: We thank the reviewer for this comment. According to your suggestion, we revised all of the above-mentioned descriptions in our manuscript (Line 138-141, 151, 158-159, 162, 169-170).
Line 143 - add a space between "14" and "years"
Response: We thank you for your pointing out. According to your suggestion, we added space between "14" and "years" (line 143).
Line 155 - remove "later"
Response: We thank the reviewer for this advice. We deleted the description "later" (Line 154).
Line 166 - remove "later"
Response: We thank the reviewer for this advice. We deleted the description "later" (Line 165).
Line 191 - remove "would" and change "decline" to "declined"
Response: We thank you for your pointing out. According to your suggestion, the description “would decline” was changed to “declined” (line 191).
Line 193 - add space between "3" and month
Response: We thank you for your pointing out. According to your suggestion, we added space between "3" and "months" (line 193).
Line 193 - replace "3" with "three"
Response: We thank you for your pointing out. According to your suggestion, the description “3” was changed to “three” (line 193).
Line 238 - replace "part" with "proportion"
Response: We thank you for your pointing out. According to your suggestion, the description “part” was changed to “proportion” (line 238).
Line 238 and 242 - Regarding "HBV cure": Complete HBV cure is currently not achievable, I would use the would HBV functional cure or HBsAg seroconversion.
Response: We thank you for your pointing out. According to your suggestion, the description “HBV cure” was changed to “HBV functional cure” (line 238-239, 242).
Line 244 - remove "great"
Response: We thank the reviewer for this advice. We deleted the description "great" (Line 244).
Lines 246-250 - Better if "A study by Iser et al. showed that among HIV-HBV coinfected patients on ART, a significant increase in the number of peripheral circulating CD4+ T cells but not in the liver. Intrahepatic changes by liver biopsy showed no significant changes..."
Response: We thank the reviewer for this comment. According to your suggestion, the description “David M Iser et al.'s report of post-ART in HIV-HBV coinfected patients showed a significant increase in the number of circulating CD4+ T cells, but reports of intrahepatic changes by liver biopsy showed no significant changes in the number of CD4+ and CD8+ T cells or T cell activation, suggesting a low correlation with HBs antigen clearance [11].” was changed to “A study of Iser et al. showed that among HIV-HBV coinfected patients on ART, a significant increase in the number of peripheral circulating CD4+ T cells but not in the liver. Intrahepatic changes by liver biopsy showed no significant changes in the number of CD4+ and CD8+ T cells or T cell activation, suggesting a low correlation with HBs antigen clearance [11].” (Line 246-250).
Line 283 - simplify "HIV and HBV" to "HIV-HBV" or "HIV/HBV"
Response: We thank you for your pointing out. According to your suggestion, the description “HIV and HBV” was changed to “HIV/HBV” (line 283).
Again, we wish to express our appreciation to the reviewers for their insightful comments on our paper.

Reviewer 2 Report
I thank the authors for their adequate responses. A bit disappointed that they left the case vingettes and did not include a more general discussion of HBsAg-loss in the HIV-HBV co-infected population, but this does not dismerit their contribution.
Some small issues:
- Figure 3. Unless there is something I do not understand, why are the bands not around the top of the bars?
- line 244. "There are some great studies looking..." is a bit informal and should be changed to "There have been studies looking..."
Author Response
We thank the reviewers for their insightful comments and suggestions, which have helped to improve our paper.
Reviewer 2
I thank the authors for their adequate responses. A bit disappointed that they left the case vingettes and did not include a more general discussion of HBsAg-loss in the HIV-HBV co-infected population, but this does not dismerit their contribution.
Response: Thank you for your kind comments.
Some small issues:
- Figure 3. Unless there is something I do not understand, why are the bands not around the top of the bars?
Response: We thank the reviewer for helpful comment. There was a technical error in displaying the standard deviation. We corrected the figures expressing as means ± standard deviation in our manuscript (Figure 3, Figure 4, Figure S1-S2).
- line 244. "There are some great studies looking..." is a bit informal and should be changed to "There have been studies looking..."
Response: We thank you for your pointing out. According to your suggestion, the description "There are some great studies" was changed to "There have been studies" (line 244).
Again, we wish to express our appreciation to the reviewers for their insightful comments on our paper.
